# A Novel Dressing Composed of Adipose Stem Cells and Decellularized Wharton’s Jelly Facilitated Wound Healing and Relieved Lymphedema by Enhancing Angiogenesis and Lymphangiogenesis in a Rat Model

**DOI:** 10.3390/jfb14020104

**Published:** 2023-02-14

**Authors:** Jen-Her Lu, Kai Hsia, Chih-Kuan Su, Yi-Hsiang Pan, Hsu Ma, Shih-Hwa Chiou, Chih-Hsun Lin

**Affiliations:** 1Department of Pediatrics, Taipei Veterans General Hospital, Taipei 11217, Taiwan; 2Section of Pediatric Cardiology, Department of Pediatrics, Taipei Medical University Hospital, Taipei 11031, Taiwan; 3Department of Medical Research, Taipei Veterans General Hospital, Taipei 11217, Taiwan; 4Division of Plastic and Reconstructive Surgery, Department of Surgery, Taipei Veterans General Hospital, Taipei 11217, Taiwan; 5Department of Surgery, School of Medicine, National Yang Ming Chiao Tung University, Taipei 11221, Taiwan; 6Institute of Pharmacology, National Yang Ming Chiao Tung University, Taipei 300093, Taiwan

**Keywords:** lymphedema, decellularized Wharton’s jelly, rat adipose stem cell, angiogenesis, lymphangiogenesis, wound-healing

## Abstract

Lymphedema causes tissue swelling due to the accumulation of lymphatic fluid in the tissue, which delays the process of wound-healing. Developing effective treatment options of lymphedema is still an urgent issue. In this study, we aim to fabricate tissue-engineered moist wound dressings with adipose stem cells (ASCs) and decellularized Wharton’s jelly (dWJ) from the human umbilical cord in order to ameliorate lymphedema. Rat ASCs were proliferated and an apparent layer was observed on dWJ at day 7 and 14. A rat tail lymphedema model was developed to evaluate the efficacy of the treatment. Approximately 1 cm of skin near the base of the rat tail was circularly excised. The wounds were treated by secondary healing (control) (*n* = 5), decellularized Wharton’s jelly (*n* = 5) and ASC-seeded dWJ (*n* = 5). The wound-healing rate and the tail volume were recorded once a week from week one to week five. Angiogenesis and lymphangiogenesis were assessed by immunochemistry staining with anti-CD31 and anti-LYVE1. The results showed that the wound-healing rate was faster and the tail volume was lesser in the ASC-seeded dWJ group than in the control group. More CD31+ and LYVE-1+ cells were observed at the wound-healing area in the ASC-seeded dWJ group than in the control group. This proves that tissue-engineered moist wound dressings can accelerate wound-healing and reduce lymphedema by promoting angiogenesis and lymphangiogenesis.

## 1. Introduction

Lymphedema is caused by the dysfunction of either the primary or secondary subcutaneous lymphatic channels. Primary lymphedema develops because of an inherited abnormality of the lymphatic system, whereas secondary lymphedema occurs after trauma or injury [1]. Secondary lymphedema is most commonly associated with lymphatic filariasis in developing countries.

In developed countries, the most common causes are the disruption of the lymphatic channels after regional lymphadenectomy or radiation therapy in cancer patients [2]. Disrupted lymphatic vessels can lead to the reflux of lymphatic fluid into the interstitial space [3]. Static lymphatic fluid can provoke a localized chronic inflammatory process, followed by abnormal extracellular matrix remodeling, thus leading to tissue fibrosis, ultimately destructing the lymphatic vessel lumen [4]. The inflammatory process around the lymphatic vessels, including abnormal enzyme function and immune cell action, further leads to a decrease in lymphatic duct contractility and decreased collateral lymphatic vessel formation [5,6,7]. This vicious cycle results in progressive tissue inflammation and fibrosis. Therefore, lymphedema delays wound-healing due to its pathophysiological and immunological effects [8,9]. Clinical manifestations include heaviness, pain, recurrent cellulitis, tissue fibrosis, fat deposition, body disfiguration and impaired quality of life [10].

Currently, therapies for lymphedema include non-surgical and surgical treatments. The treatment goal is to alleviate symptoms, reduce swelling, prevent progression and reduce the risk of infection [11]. Non-surgical treatments mainly encompass physical therapy by manual decompression or mechanically assisted compression to improve lymph fluid return [12]. Surgical treatments are divided two categories: excision and physiological restoration. Excision procedures encompass debulking, suction-assisted lipectomy or direct excisional procedures for pathological tissues [13]. The physiological restoration of lymph fluid return includes lymph duct transplantation, lymph-venous bypass or lymph node transplantation [14]. However, the greatest improvements have been reported with combined treatment programs. Evidence-based outcomes are still needed [12]. Thus, there is still space for the investigation of new therapies for lymphedema.

Recently, adipose stem cells have also gained attention in regard to the treatment of secondary lymphedema [15]. Some preclinical and clinical studies have already revealed promising results for adipose stem cell therapy in the treatment of secondary lymphedema [15]. Previous reports have revealed the role and efficacy of ASCs in the improvement of tissue regeneration, especially in soft tissue reconstruction, such as breast reconstruction, wound-healing and the treatment of scars and soft tissue defects [16,17,18,19]. Adipose stem cells may have clinical potential due to advantages such as being low risk and high yield, along with having numerous cell types [20]. The possible mechanism of adipose stem cells on lymphedema is not only in their assistance of lymphangiogenesis but also in their anti-inflammatory effect. Adipose stem cells and their secretory factors can significantly promote the proliferation, migration and tube formation of lymphatic endothelial cells [21,22]. Active T cells play an important role in the progressive development of lymphedema [23,24]. The infiltration of lymphocytes and macrophages is also observed in the microenvironment of lymphedema [25,26]. In addition, T cell-derived cytokines may negatively regulate lymphangiogenesis [23]. Theoretically, the factors secreted by adipose stem cells have anti-inflammatory, anti-fibrosis and immune modulation abilities and could be a resource to reverse T cell-related chronic inflammation in lymphedema [20].

The human umbilical cord is receiving increasing attention due to its abundant content of mesenchymal stem cells, which have been employed in preclinical and clinical trials [27,28]. The structure of the umbilical cord consists of the outer amnion epithelium, Wharton’s jelly and three blood vessels [29]. Wharton’s jelly is mucoid connective tissue comprising numerous growth factors (IGFBP 1, 2, 3, 4 and 6; TGF-α; and PDGF-AA), cytokines (RANTES, IL-6R and IL-16, MCSFR, MIP-1a, TNF-RI, TNF-RII, IL-1RA, TIMP-1, TIMP-2, ICAM-1, G-CSF and GDF-15) and extracellular matrix (hyaluronic acid, collagen and glycosaminoglycans) [29,30,31,32]. Wharton’s jelly plays a crucial antibacterial role, reducing inflammation and improving wound-healing and cell homing. In order to avoid adverse reactions of the host, a decellularization technique was employed. dWJ has been reported to provide a biocompatible structure for cartilage [33], hepatic tissue [34], neural tissue [33] and wound repair and regeneration [35]. Decellularized Wharton’s jelly not only has the advantages of easy acquisition and cost effectiveness, it is also suitable for stem cell adhesion and colonization. Therefore, it has great potential for tissue engineering.

Through cell therapy, the application of tissue engineering techniques in the treatment of secondary lymphedema could present an alternative treatment method, but there are few studies on this subject. Tissue engineering incorporates cells, scaffolding and signals to enhance tissue repair and regeneration [36]. Currently, tissue engineering approaches for lymphatic vessel regeneration include cell-seeded scaffolds for the ex vivo assembly of lymphatic grafts and the in situ assembly of lymphatic structures for in vivo development. Advances in scaffolding techniques can guide engineering efforts to generate large lymphatic vessels [37,38]. In addition to the hydrogel matrix, the acellular dermal matrix has also shown potential in promoting lymphangiogenesis in wounds [39]. Thus, we aim to find an effective method to treat lymphedema. We hypothesize that adipose-stem-cell-seeded dWJ could promote lymphangiogenesis after implantation in a rat tail lymphedema model. The relief of tail edema and wound-healing were evaluated in terms of efficacy compared to the healing of the secondary and scaffolding-only groups. 

## 2. Material and Methods

### 2.1. Rat Tail Lymphedema Model

The study was approved by the Institutional Animal Care and Use Committee of Taipei Veterans General Hospital. All animal care complied with the Guide for the Care and Use of Laboratory Animals (No. IACUC 2019-108, IACUC 2020-155, IACUC 2020-240). Fifteen male Sprague Dawley rats ((BioLASCO, Yilan, Taiwan)) aged 20 to 24 weeks were used for this research. Animals were divided into three groups randomly, which were as follows: (i) secondary healing (control) (*n* = 5); (ii) decellularized Wharton’s jelly (dWJ) (*n* = 5); and (iii) ASC-decellularized Wharton’s jelly (ASC/dWJ) (*n* = 5). Sprague Dawley rats were anesthetized with an intraperitoneal injection of 50 mg/kg body weight of Zoletil 50 (Virbac, Carros cedex, France). The anesthetized rats were placed in a prone position over a warm pad. The diameter of the tail and the excision wound was calculated by a Vernier scale. Evan’s blue dye (E2129, Sigma-Aldrich, St. Louis, MO, USA) was injected into the tail base before operation in order to identify the lymph vessels. After shaving and sterilization, a circumferential excision (1 cm in width) of skin at a 5 mm distance from the base of the tail was made (Figure 1A). The superficial lymphatic network was removed, and the deep lymphatic system and lateral tail veins were preserved (Figure 1B). The rats recovered from anesthesia in a separate cage after the operation and received food and water ad libitum. 

Tail volume and wound width for each group were evaluated to track tail lymphedema and wound-healing. The data for each group were compared from week 1 to week 5 after surgery. The tail volumes were calculated using a truncated cone formula according to previously published methods (Figure 1C) [40]. For this purpose, tail diameter at 2 cm and 16 cm from the tail’s end were measured by two blinded investigators using a digital caliper. The increase in volume, which represents lymphedema, was defined as post- vs. pre-operative volumes of the same animal for each week. 

The digital images of the wound were captured weekly in order to quantify the wound width. Width was determined by the average of three locations (the edge of the wound on both sides and the center of the wound). The values were measured from digital pictures using ImageJ software (version 1.53t).

### 2.2. Isolation and Characterization of Rat Adipose Stem Cells (rASCs)

Three male Sprague Dawley rats (BioLASCO, Yilan, Taiwan) aged 8 to 10 weeks were assigned as fat tissue donors. Rat adipose stem cells (rASCs) were harvested from the abdomens and the inguinal fat pads. The fat tissue was immediately placed in serum-free DMEM medium containing 500 U/mL penicillin streptomycin after harvest. The samples were then cut into 2 mm^3^ pieces in DMEM medium (high-glucose, serum-free; Life Technologies, Grand Island, NY, USA) containing 1 mg/mL collagenase (Sigma-Aldrich, St. Louis, MO, USA) and digested at 37 °C for 2–5 h. The mixture was sieved through a 100 μm mesh to eliminate debris and then centrifuged at 300× *g* for 10 min. The pellet was then resuspended in complete medium (high-glucose DMEM, 10% FBS, 1% Penicilin/Streptomycin) and cultured overnight under 5% CO_2_ at 37 °C. The medium was changed to high-glucose DMEM, 10% FBS, 1% P/S, 2 mM *N*-acetyl-l-cysteine (NAC; Sigma A8199) and 0.2 mM L-ascorbic acid-2 phosphate (Asc 2P; Sigma A8960) every other day from then on until there was no oil residue and the cells had reached 80% confluence. The cells of passages seven to ten were characterized with antibody CD29 (102208, 1.5:1000, BioLegend, San Diego, CA, USA), CD90 (202524, 1.5:1000, BioLegend), CD31 (NB100-04796, 2.5:100, Novus Biologicals), CD11b/c (201805, 1:100, BioLegend) and CD45 (202205, 1:100, BioLegend) staining by FASC (BD Caliber, USA) and used in the following study. The adipogenic, osteogenic and chondrogenic potential of rASCs were measured. After using STEMPRO differentiation kit (Gibco, Life Technologies, Carlsbad, CA, USA), according to the manufacturer’s protocols, the adipogenic, osteogenic and chondrogenic outcomes were stained with Oil Red O (Sigma-Aldrich, St. Louis, MO, USA), Alizarin Red (Sigma-Aldrich) and Alcian Blue (Sigma-Aldrich), respectively.

### 2.3. Preparation of Decellularized Wharton’s Jelly

Human umbilical cords were obtained with informed patient consent and approval from the Institutional Review Board of Taipei Veterans General Hospital (2013-08-020BC). One umbilical cord from a healthy, full-term birth donor was utilized in this experiment. After the two umbilical arteries were removed using blunt dissection, the umbilical vein was cut longitudinally in order to peel off the vessel wall. The remaining Wharton’s jelly was approximately 3 cm wide when spreading out. It was divided into three segments of 5~8 cm in length for decellularization. Decellularization was performed in four steps by merging the samples in 0.1% SDS, phosphate-buffer saline (PBS) and medium 199 containing 12% FBS and PBS, sequentially. Each step was performed under sterile conditions with 100 rpm shaking at 37 °C for two days. The cellularity and histomorphology of dWJ were examined using hematoxylin and eosin (HE; Sigma Aldrich) and DAPI staining.

### 2.4. Survival and Proliferation Assay of rASCs 

The survival and proliferation of rASCs on the Petri dishes and decellularized Wharton’s jelly were measured by CCK-8 assay, according to the manufacturer’s instructions. In brief, the cells were harvested and directly seeded in a 24-well plate or on 2 cm^2^ decellularized Wharton’s jelly at a density of 1 × 10^6^ per well in 2 mL medium. The cells were cultured at 37 °C and 5% CO_2_, and the medium was replaced every two days. The cells were further assessed with a CCK-8 assay at days 1, 3, 5, 7, 9, 11, 14 and 17. The absorbance of each well was measured at 450 nm with a reference wavelength of 655 nm. Each assay was performed in triplicate.

### 2.5. Evaluation of the Effect of ASC-Seeded Decellularized Wharton’s Jelly on Tail Edema and Wound-Healing

Decellularized Wharton’s jelly was sectioned into an area of 1.5 × 3 cm^2^. A total of 1 × 10^6^ rat ASCs/cm^2^ was seeded onto the dWJ at a density of 1 × 10^6^ /cm^2^ in 2 mL culture medium per well of a 24 well plate and incubated at 37 °C and 5% CO_2_ for 30 min. Fresh medium was changed every two days for 14 days. The scaffolds with and without rASCs were applied to the circumferential excision site of the rat tail lymphedema model by suturing with 3–0 nylon and covered by 3M™ Tegaderm for a week. Wound width and tail volume were recorded and calculated every week for 5 weeks. The rats were sacrificed, and the wound areas were harvested. All samples were fixed in 10% (*v/v*) neutral-buffered formalin solution (Leica, Wetzlar, Germany) for 24 h, processed paraffin blocked and sectioned by 5 μm thickness. Immunohistochemistry staining was performed to evaluate angiogenesis and lymphangiogenesis with anti-CD31 (Abcam, Cambridge, UK) and anti-LYVE1 (Invitrogen, Carlsbad, CA, USA). Immunohistochemistry (IHC) was performed on the Bond-MAX system (Leica Biosystems, Melbourne, Australia), which was an automated IHC staining system with the Bond Polymer Refine Detection Kit (Leica, Deer Park, IL, USA), according to the manufacture’s instruction. The tissue sections were pretreated using heat-mediated antigen retrieval with sodium citrate buffer (pH 6, epitope retrieval solution 1) for 30 min. Then, the samples were incubated with anti-CD31 and anti-LYVE1 at 1:100 dilution for 60 min at room temperature, followed by secondary HRP antibodies. After blocking with peroxide for 5 min, DAB was applied as the chromogen on the stained sections. Finally, hematoxylin for nuclear counter staining was processed and slides were mounted with DPX (Sigma-Aldrich, St. Louis, MO, USA). Images were captured under the bright field of a ZEISS inverted microscope (Zeiss, Jena, Germany).

### 2.6. Statistical Analysis 

All data are shown as mean ± SD. The Mann–Whitney U test was applied to compare the two experimental groups. The Kruskal–Wallis test with post hoc Mann–Whitney U test was used to compare data for three groups. *p* < 0.05 was considered significant. IBM SPSS Statistics 19 (version 19; SPSS, Chicago, IL, USA) was used. 

## 3. Results

### 3.1. Characteristic of Decellularized Wharton’s Jelly and Rat ASCs 

The characterization of the rat ASCs was evaluated by FACS. The evaluation indicated that 98.9% of CD29 and 97.9% of CD90 were expressed, while less than 5% of CD11b/c, CD31 and D45 was observed on the cells (Figure 2A). The differentiation assays of ASCs toward adipocytes, chondrocytes and osteocytes were successful (Figure 2B–D). These results confirmed that the cells were rat ASCs. 

The decellularized efficacy of Wharton’s jelly was evaluated by HE and DAPI staining. The HE staining indicated that the components of cells were absent from dWJ, whereas the structure was intact (Figure 1E), similar to fresh Wharton’s jelly (Figure 1F). DAPI staining further confirmed that the cell nuclei was removed (data not shown).

### 3.2. Rat ASCs Expanded on Decellularized Wharton’s Jelly

A CCK-8 assay was used to assess the growth rate of rASCs on both culture dishes and dWJ at day 1, 3, 5, 7, 9, 11, 14 and 17 of cultivation. We observed that the trend of the growth curve of rASCs on dWJ was similar to that of the 24-well plate, although the absorbance at 455–650 nm was slightly less (*n* = 3, Figure 2E). The grafts of rASCs seeded on dWJ for 7 and 14 days were harvested for HE and DAPI staining. The number of cells generated on dWJ increased over time. This demonstrates good attachment and proves that rASCs can grow on dWJ, as shown in Figure 2F.

### 3.3. The Effect of ASC/dWJ on Rat Tail Model

After the wound in the rat tail was made, dWJ (*n* = 5) and ASC/dWJ (*n* = 5) were applied to the trauma site. An untreated group acted as a secondary dressing control (*n* = 5). It was observed that the most significant edema occurred near the wound. The wound healed from the third week and had gradually closed by the fifth week (Figure 3). Tail lymphedema presented as the increased fold of tail volume compared with week 0. The lymphedema in control group was swollen in the first week after surgery, which decreased slightly in the second week; then, the curve rose in the third week to the level of the first week and was stable to the end. In the dWJ group, the caudal expansion trend was similar to that of the control group but the expansion was slightly less. For the secondary healing control group, the values for the increase in lymphedema from week 1 to week 5 were 1.4 ± 0.32, 0.32 ± 0.29, 1.36 ± 0.28, 1.53 ± 0.16 and 1.49 ± 0.20. The values of the dWJ group were 1.27 ± 0.27, 1.22 ± 0.15, 1.26 ± 0.24, 1.34 ± 0.30 and 1.24 ± 0.23, and those of the ASC/dWJ group were 1.23 ± 0.15, 1.28 ± 0.25, 1.16 ± 0.07 and 1.10 ± 0.10. The ASC/dWJ group showed a significant change in comparison to the secondary dressing control group regarding the consequent anti-edema at weeks 4 and 5 (*p* < 0.05), while the average of the dWJ group was smaller than control and larger than the ASC/dWJ group but had no significant difference due the large variation between the animals (Figure 4A). The width of the wound was used to estimate the speed of wound-healing. As shown in Figure 4B, the width of the control group (1.21 ± 0.25) was statistically greater than the other two test groups at week 1, but the latter two groups had no difference (0.99 ± 0.13 vs. 0.99 ± 0.14). At week 5, although the size of the wound in the ASC/dWJ group (0.10 ± 0.05 cm) was smaller than the dWJ (0.21 ± 0.07 cm) and control group (0.27 ± 0.12 cm), there was no significance because the value was too small to be measured. Regarding wound development, the control group was larger in the first week after surgery and continued shrinking over the next two to four weeks; the wounds in the dWJ and rASC/dWJ groups continued to decrease after surgery without enlargement in the early stage (Figure 4B).

It was observed that the most significant edema occurred near the wound. The wound healed from the third week and had gradually closed by the fifth week.

### 3.4. Histological Outcome

Five weeks after surgery, subcutaneous CD31 and LYVE-1 IHC staining results were obtained. It was observed that LYVE-1^+^ cells formed the lumen of the lymphatic vessels, and there were also free cells in the interstitial space (Figure 5). However, the staining results showed that the luminal endothelial cells exhibited both LYVE-1 and CD31 (Figure 5A–C), which was different from past studies. From the calculation data, it can be seen that the number of cells exhibiting CD31 and LYVE-1 increased in both the dWJ dressing group and the rASC/dWJ group compared to the rASC/dWJ group. There was no significate difference between the rASC/dWJ group and control. Nevertheless, in the rASC/dWJ group, there were statistically significant increases in comparison to the control and dWJ groups (*p* < 0.05) (Figure 5D,E).

## 4. Discussion

There are different preclinical models for lymphedema research, included murine, rabbit, canine and porcine hindlimb and tail. The surgical procedures performed mainly included circumferential excision, with or without lymphadenectomy and radiation [25,41,42,43,44,45,46,47,48,49,50,51]. Theoretically, the simulation of human cancer therapeutics with lymphadenectomy and irradiation was close to the scenario of secondary lymphedema. In our study, we established a rat tail lymphedema model by the circumcision of skin and disruption of lymph ducts. The subcutaneous vascular bundles were preserved to avoid skin necrosis. An increase in the circumference and volume of the tail distal to the wound was observed. The advantage of the current model is that it is simple and fast, with a reliable outcome. We observed that the wound after circumcision excision healed in rats in about five weeks, which is similar to the results of other studies [52,53]. Compared with the wounds of dWJ and ASC/dWJ group, those of control group were more than one cm wide after one week post-operation. This initial wound enlargement following skin excision was due to muscular retraction and lymphedema. However, the edematous status of the tail in the experimental groups decreased gradually one week after excision and persisted up to five weeks. This finding is slightly different from mouse models, which showed that after tail excision, gradual tail volume increased over five weeks [54]. This difference could be due to the different injury responses of different animal species. Another cause could be calculation error because edematous part of the tail is completely cone-shaped. 

Several studies have suggested the relevance of insufficient lymphangiogenesis to impairing wound-healing processes [55,56,57,58,59,60]. Lymphatics have been associated with granulation tissue formation, matrix remodeling and leukocyte trafficking in wound-healing [61]. Lymphangiogenesis is considered to play a role in the late phase of wound-healing [62]. Molecular signals in tissue injury, such as high-mobility group box 1 (HMGB1) and heat shock protein (HSP) 70, are closely related to local tissue lymphatic fluid stasis [53]. The advantage of the murine tail circumcision model is that not only can it evaluate the therapeutic effect on lymphedema, it also allows the observation of the wound-healing process under intervention. 

Biomaterials from extracellular matrices are innately bioactive and biodegradable and considered as surface ligands for cell adhesion [35]. Natural materials have shown benefits in promoting wound-healing in regard to angiogenesis and lymphangiogenesis. The matrices allow the infiltration of vascular and lymphatic vessels [63]. Well-structured morphological and functional development in vascular and lymphatic endothelial structures could be observed in the wound repaired by acellular dermal matrix [36]. Wharton’s jelly is derived from connective tissue surrounding umbilical cord vessels and is rich in collagen, hyaluronan and glycosaminoglycans (GAGs) [31]. It also contains abundant peptide growth factors, such as the insulin-like growth factor-1 (IGF-1) and the platelet-derived growth factor (PDGF) [32]. These biological characteristics promote cell adhesion, migration and proliferation [64]. Decellularized Wharton’s jelly has been investigated in wound-healing and tracheal and vocal cord repair [61,65,66]. The microenvironment of dWJ has been demonstrated to support the viability and function of stem cells [34,67,68]. In our results, adipose stem cells were able to attach and proliferate on dWJ from day 7 and cells increased in a 14 day culture. In vivo results also showed that ASC-seeded dWJ was most effective in reducing lymphedema in rat tails. Therefore, decellularized Wharton’s jelly could be an ideal carrier for stem cell implantation in regenerative medicine applications.

“Lymphvasculogenesis” indicates the de novo incorporation of the precursors of non-venous origin [69]. This may support the use of cell therapy for lymph vessel regeneration. Stem cells have shown the capacity to differentiate to lymphatic endothelial cell lineages, allowing them to serve as a delivery system to locally and durably release growth factors for angiogenesis or lymphangiogenesis [50]. Adipose stem cells have been induced to form lymphatic endothelial cells using a medium containing VEGF-C, bFGF and other growth factors [70]. The paracrine effect of adipose stem cells can maintain lymphatic endothelial cell morphology, survival and induction to form a vessel-like structure [21,71]. The use of optimal biomaterials as carriers can retain five to six times more cells compared to saline immediately following transplantation, overcoming stem cell loss due to injection [72,73]. Decellularized Wharton’s jelly meets the needs of stem cell growth [74,75]. Besides lymphangiogenesis, angiogenesis was crucial for accelerating the rate and quality of the wound-healing process by providing nutrition to the injury [76]. As shown by the use of decellularized biological material with stem cells in our study, the clinical applications of biomaterial-assisted cell therapies may hold great promise in regenerative therapy for wound-healing and lymphedema treatments in the future. 

There are limitations to our study. First, we only used one lymphatic marker and one endothelial marker to define the lymphatic and vascular endothelial cells. Other markers of lymphatic microvasculature, such as secondary lymphoid chemokine (SLC), VEGF receptor-3 (VEGFR-3) and podoplanin, and more endothelial markers of blood vessel, including the von Willebrand factor (VWF) and endothelial nitric oxide synthase (eNOS), might have been included to enhance the evidence that ASCs/dWJ dressings were precisely recruited the lymphatic and blood vessel endothelial cells. Second, we did not explore whether ASCs/dWJ dressings influenced the secretion of pro- and anti-lymphangiogenic cytokines, such as interleukin- 8 (IL-8) and transforming growth factor-β1 (TGF-β1), in the neogenesis of lymphatic endothelial cells. More experiments are required to confirm and strengthen these results. Finally, LYVE-1 and CD31 are generally considered to be expressed in the lymphatic endothelium and vascular endothelium, respectively. However, we found that luminal endothelial cells express both LYVE-1 and CD31. The CD31+ and LYVE-1+ areas were fairly close during IHC staining, which suggests that the capillary and microlymphtic vessel were concomitant and assembled a vascular network during the wound-healing process. Research has shown that LYVE-1 and CD31 are expressed in the yolk sac microvascular plexus, pulmonary vascular endothelial cells and the endocardia layer during the embryonic period in rats [77]. It has also been shown that both LYVE-1 and CD31 are observed in the masseter muscle of post-natal rats but with different levels of expression. However, there is no literature verifying this phenomenon in the caudal cortex of rats, so further investigation is needed. In addition, although the results indicate that functional recovery has occurred via rASC/dWJ treatment, physiologic functional investigation, such as ICG or Patent Blue dying, is still necessary and the relative tests will be included in future studies.

## 5. Conclusions

The tissue-engineered moist wound dressings fabricated with ASCs and decellularized Wharton jelly could attenuate lymphedema by promoting lymphatic vessel and capillary formation in a rat tail lymphedema model. It further proved that wound-healing could be accelerated via restoring blood and lymphatic circulation after skin injury. Therefore, this dressing has potential as an advanced dressing for further clinical application. 

## Figures and Tables

**Figure 1 jfb-14-00104-f001:**
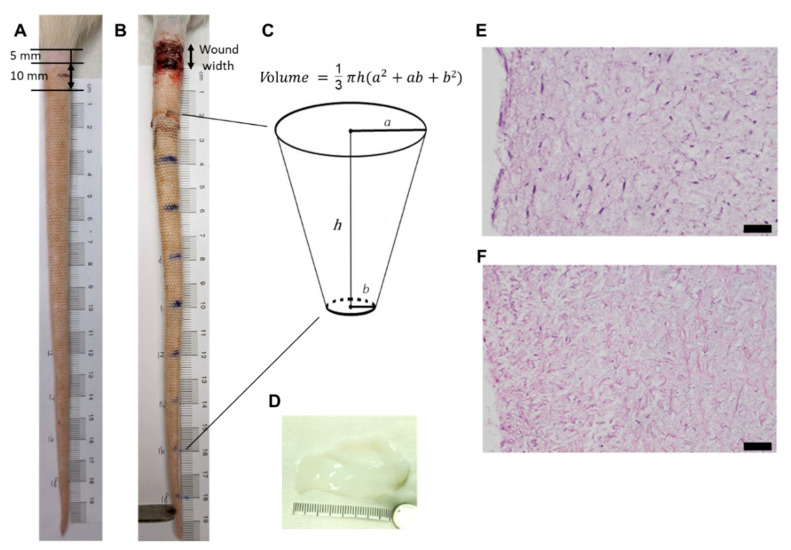
Rat tail lymphedema model and decellularized Wharton’s jelly from human umbilical cord. (**A**) rat tail before operation demonstrating the wound position and size; (**B**) rat tail after operation showed the lymphedema distal to the wound; (**C**) calculation of rat tail volume; (**D**) gross view of the decellularized Wharton’s jelly; (**E**) HE staining of Wharton’s jelly from human umbilical cord; (**F**) HE staining of decellularized Wharton’s jelly from human umbilical cord. The figure shows that the cells were eliminated from the Wharton’s jelly. Magnification 200×; scale bar = 5 μm.

**Figure 2 jfb-14-00104-f002:**
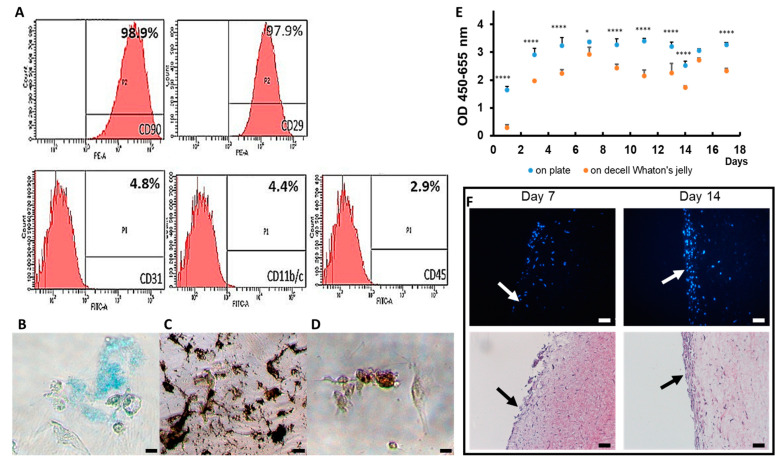
Rat ASC. (**A**) in vitro characterization of ASC markers at passage 7 to 10. The image and characterization of ASCs by FACS; (**B**) the chondrogenic differentiation of ASC by Alcian Blue staing; (**C**) the osteogenic differentiation of ASC by Alizarin Red staining; (**D**) the adipogenic differentiation of ASC by Oil Red O staining; (**E**) the proliferation curve of ASCs on the Petri dish and dWJ; (**F**) DAPI and HE staining of ASC-dWSJ after cells were seeded for 7 days and 14 days. Magnification 200×; scale bar = 5 μm. * *p* < 0.05, **** *p* < 0.001.

**Figure 3 jfb-14-00104-f003:**
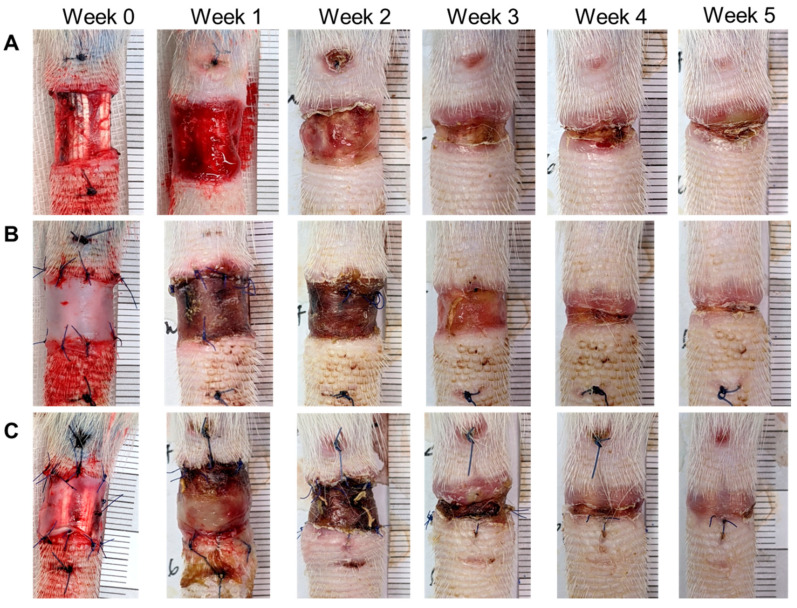
Images of rat tail lymphedema and wound-healing after dWJ and rASC/dWJ treatment for five weeks after injury. (**A**) images of rat tails of control group from week 0 to week 5; (**B**) images of rat tails of dWJ group from week 0 to week 5; (**C**) images of rat tails of dWJ group from week 0 to week 5. Photographs of week 0 represent the mouse tails treated without and with dressings just after surgical excision and the wound width for each group was made as similar as possible. During wound development, the lymphedema of control group was the most significant compared with the two treatment groups. The wound widths of rASC/dWJ groups seem smaller than the other two groups from week one to five after treatment. *n* = 5/group.

**Figure 4 jfb-14-00104-f004:**
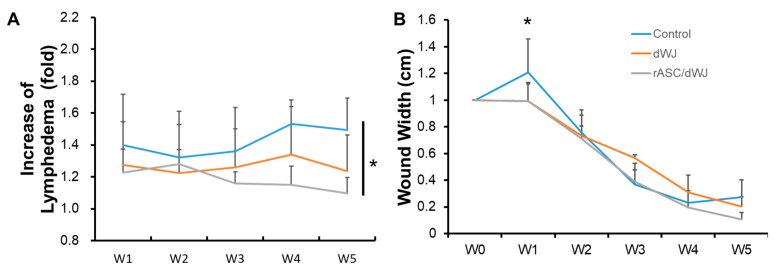
Evaluation of lymphedema and wound-healing after treatment with dWJ and rASC/dWJ for 5 weeks. (**A**) increase in lymphedema was assessed by tail volume for each week compared with pre-operational tail volume; (**B**) wound width. * *p* < 0.05.

**Figure 5 jfb-14-00104-f005:**
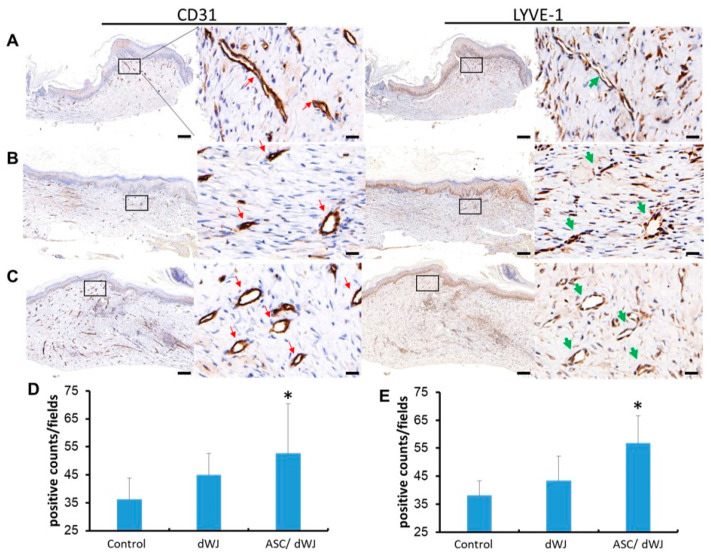
The angio-expression of CD31 and LYVE-1 in the excision sites after five weeks. Immunohistochemistry of (**A**) control group; (**B**) dWJ group; (**C**) rASC/dWJ group. The results showed that although both CD31 and LYVE-1 positive cells in treating groups (rASC/dWJ group and dWJ group) were more than in the control group, the rASC/dWJ group contained the most positive cells among these three groups. The red arrows point to capillary blood vessels and the green arrows to lymphatic vessels. Magnification of 40×, scale bar = 200 μm; magnification of 400×, scale bar =20 μm. (**D**) The qualification of CD31 positive cells; (**E**) the qualification of LYVE-1 positive cells. The statistical data demonstrate that both CD31 and LYVE-1 positive cells in rASC/dWJ group were significantly more than the control and dWJ group. * *p* < 0.05.

## Data Availability

Not applicable.

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
