# Peer review of "A Novel Dressing Composed of Adipose Stem Cells and Decellularized Wharton’s Jelly Facilitated Wound Healing and Relieved Lymphedema by Enhancing Angiogenesis and Lymphangiogenesis in a Rat Model"

_jfb, 2023, doi:10.3390/jfb14020104_

Round 1
Reviewer 1 Report
REVIEW REPORT ON THE MANUSCRIPT JFB-2130345 TITLED: A NOVEL DRESSING COMPOSED OF ADIPOSE STEM CELLS AND DECELLUARIZED WHARTON’S JELLY FACILITATED WOUND HEALING AND RELIEF OF 3 LYMPHEDEMA BY ENHANCING ANGIOGENESIS AND LYMPHANGIOGENESIS IN A RAT MODEL
GENERAL
The study describes some therapeutic approaches to lymphedema in a rat model using a novel dressing of decellularized Wharton’s jelly (dwj) and adipose tissue stem cells (ASC). Wound healing rate was reported to be faster and the tail volume was lesser in the ASC-seeded dWJ group than in the control group. Angiogenesis and lymphangiogenesis were respectively assessed by More CD31+ and LYVE-1 + cells were observed at the wound healing area in the 31 ASC-seeded dWJ group than in the control group
INTRODUCTION
This section is well articulated apart from a few grammatical errors,
MATERIAL AND METHODS
Line 132-reference to tail lymphoma is misleading as this would indicate there was cancer.
Section 2.2: isolation of ASCs line 143:
How many rats did you use in this part of the experiment? What types of rats were these? Were they juvenile or adult rats, male or female?
2.3. Preparation of Decellularized Wharton’s Jelly – LINE 161- HOW MANY umbilical cords and what were their gestation ages? How exactly were the cords dissected out?
2.5. Evaluation of the effect of ASC-seeded dWJ on tail edema and wound healing - LINE181
Do not start statements with abbreviations or digits.
Briefly describe histological processing.
RESULTS
The figure captions in this section are not informative. The authors should clearly indicate what the figure is showing.
DISCUSSION
Many ambiguous statements in this section need rephrasing. Do not open a sentence with a digit or abbreviation.
More
REFERENCES
Some references (30 & 31) are missing from the list.
NB: more comments are to be found in the attached tracked-in PDF copy
Author Response
Q1. This INTRODUCTION section is well articulated apart from a few grammatical errors,
Answer: Thanks for your indication.
We corrected in Line 49 ~51.
MATERIAL AND METHODS
Q2. Line 132-reference to tail lymphoma is misleading as this would indicate there was cancer.
Answer: Thanks for your opinion.
It is a misspelling and has been revised to “lymphedema”. -- Line 141
Q3. Section 2.2: isolation of ASCs line 143:
How many rats did you use in this part of the experiment? What types of rats were these? Were they juvenile or adult rats, male or female?
Answer: Thanks for your opinion. We revised as below.
For rat ASC isolation:
Three male Sprague Dawley rats (BioLASCO, Yilan, Taiwan) age from 8 to 10 weeks were assigned as fat tissue donor. – Line 152 to 153.
For rat tail lymphedema model
Fifteen male Sprague Dawley rats ((BioLASCO, Yilan, Taiwan)) age from 20 to 24 weeks were used for this research. Animals were divided into three groups randomly which were i. Secondary healing (control) (n=5); ii. decellularized Wharton’s jelly (dWJ) (n=5) and iii. ASC- decellularized Wharton’s jelly (ASC/dWJ) (n=5). – Line 120 to 123.
Q4. 2.3. Preparation of Decellularized Wharton’s Jelly – LINE 161- HOW MANY umbilical cords and what were their gestation ages? How exactly were the cords dissected out?
Answer: Thanks for your opinion. The relative information is supplied
Human umbilical cords were obtained with informed patient consent and approval from the Institutional Review Board of Taipei Veterans General Hospital (2013-08-020BC). One umbilical cord from healthy, full-term birth donor was utilized in this experiment. After the two umbilical arteries were removed using blunt dissection, the umbilical vein was cut longitudinally in order to peel off the vessel wall. The remaining Wharton's jelly was approximately 3-cm wide when spreading out. It was divided into three segments of 5~8 cm in length for decellularization. – Line 174 to 180.
Q5. 2.5. Evaluation of the effect of ASC-seeded dWJ on tail edema and wound healing - LINE181 Do not start statements with abbreviations or digits.
Answer: Thanks for your opinion. We corrected revised according to your suggestion as below:
Evaluation of the effect of adipose stem cell-seeded decellularized Wharton’s jelly on tail edema - Line 194--195
Q6. Briefly describe histological processing.
Answer: Thanks for your opinion. We add the description as below.
The rats were sacrificed and the wound areas were harvested. All samples were fixed in10% (v/v) neutral-buffered formalin solution (Leica, USA) for 24 h, processed paraffin blocked and sectioned by 5μm thickness. Immunohistochemistry staining were preformed to evaluate angiogenesis and lymphangiogenesis with anti-CD31 (Abcam, UK,), anti-LYVE1 (Invitrogen, USA). Immunohistochemistry was performed on Bond-MAX system (Leica Biosystems, Melbourne, Australia) which was an automated IHC staining system with the Bond Polymer Refine Detection Kit (Leica, UK) according to the manufacture’s instruction. The tissue sections were pretreated using heat-mediated antigen retrieval with sodium citrate buffer (pH 6, epitope retrieval solution 1) for 30 min. Then, the samples were incubated with anti-CD31and anti-LYVE1 at 1:100 dilution for 60 min at room temperature, followed by secondary HRP antibodies. After blocking with peroxide for 5 min, DAB was applied as the chromogen on the stained sections. Finally, hematoxylin for nuclear counter staining was processed, and slides were mounted with DPX (Sigma-Aldrich, MO USA). Images were captured under the bright field of a ZEISS inverted microscope (Zeiss,Germany). – Line 202 to 216.
RESULTS
Q7. The figure captions in this section are not informative. The authors should clearly indicate what the figure is showing.
Answer: Thanks for your opinion. The explanation of Figures was amended in Line 226~228, 230~233 and 283~290.
DISCUSSION
Q8. Many ambiguous statements in this section need rephrasing. Do not open a sentence with a digit or abbreviation.
Answer: Thanks for your opinion. We revised ambiguous statements and the sentence started with a digit or abbreviation. -Line 305, 306~309, 310, 312, 334, 339, 351~356,
REFERENCES
Q9. Some references (30 & 31) are missing from the list.
Answer: Thanks for your opinion. We added the details of References of 30 & 31 which were 33 and 34 after revising as below:
[33] L. Penolazzi, M. Pozzobon, L. S. Bergamin, S. D'Agostino, R. Francescato, G. Bonaccorsi, P. De Bonis, M. Cavallo, E. Lambertini and R. Piva (2020) Extracellular Matrix From Decellularized Wharton's Jelly Improves the Behavior of Cells From Degenerated Intervertebral Disc. Front Bioeng Biotechnol. 8: 262.
[34] M. Kehtari, B. Beiki, B. Zeynali, F. S. Hosseini, F. Soleimanifar, M. Kaabi, M. Soleimani, S. E. Enderami, M. Kabiri and H. Mahboudi (2019) Decellularized Wharton's jelly extracellular matrix as a promising scaffold for promoting hepatic differentiation of human induced pluripotent stem cells. J Cell Biochem. 120: 6683-6697
Review 2
Open Review
(x) I would not like to sign my review report
( ) I would like to sign my review report
English language and style
( ) English very difficult to understand/incomprehensible
( ) Extensive editing of English language and style required
( ) Moderate English changes required
(x) English language and style are fine/minor spell check required
( ) I don't feel qualified to judge about the English language and style
|
Yes |
Can be improved |
Must be improved |
Not applicable |
|
|
Does the introduction provide sufficient background and include all relevant references? |
( ) |
(x) |
( ) |
( ) |
|
Are all the cited references relevant to the research? |
(x) |
( ) |
( ) |
( ) |
|
Is the research design appropriate? |
( ) |
( ) |
(x) |
( ) |
|
Are the methods adequately described? |
(x) |
( ) |
( ) |
( ) |
|
Are the results clearly presented? |
( ) |
( ) |
(x) |
( ) |
|
Are the conclusions supported by the results? |
( ) |
( ) |
(x) |
( ) |
Comments and Suggestions for Authors
Author concluded that ASC with decellularised Warton Jelly improved both capillary formation and lymphatic vessel foramtion. This is not enough evidence for this concluison only by CD31 and LYVE-1 IHC stainings since they were colocalised.
The author should perform addtional IHC/IF stainings to further differentiate capillaries and lympahtic vessesels to support the conlcusion.
Answer: Thanks for your opinion. Observed from Figure 5,
The CD31+ and LYVE-1+ area was quiet adjacent in the IHC staining which hints the capillary and microlymphtic vessel were concomitant and assemble a vascular network during the wound healing process.
We explained in the section of DISCUSSION ----Line366~369.
Please see the attachment in which all Revision is Written with BLUE words.

Reviewer 2 Report
Author concluded that ASC with decellularised Warton Jelly improved both capillary formation and lymphatic vessel foramtion. This is not enough evidence for this concluison only by CD31 and LYVE-1 IHC stainings since they were colocalised.
The author should perform addtional IHC/IF stainings to further differentiate capillaries and lympahtic vessesels to support the conlcusion.
Author Response
The author should perform addtional IHC/IF stainings to further differentiate capillaries and lympahtic vessesels to support the conlcusion.
Answer: Thanks for your opinion. Observed from Figure 5,
The CD31+ and LYVE-1+ area was quiet adjacent in the IHC staining which hints the capillary and microlymphtic vessel were concomitant and assemble a vascular network during the wound healing process.
We explained in the section of DISCUSSION ----Line366~369.
Please see the attachment in which all Revision is Written with BLUE words.

Reviewer 3 Report
The authors developed a wound dressing composed of decellularized Wharton's jelly (dWJ) and rat adipose stem cells (rASCs) to accelerate wound healing by enhancing lymphangiogenesis and angiogenesis in a rat tail lymphedema model. The manuscript is well-written; however, revisions are required. The manuscript should be revised in response to the comments.
1. In the "Abstract," the authors should include a background statement on the effect of lymphedema on delayed wound healing.
2. The results of in vitro tests (i.e., survival and proliferation of rASCs) should be mentioned in the "Abstract."
3. The wound healing process (for the reference: https://www.sciencedirect.com/science/article/pii/S0141813021021541), as well as the relationship between lymphedema and delayed wound healing (for the reference: https://www.liebertpub.com/doi/abs/10.1089/wound.2018.0871), should be explained in the "Introduction" section.
4. Could the authors elaborate on the evaluation of the differentiation potential of ASCs in the "Materials and methods" section?
5. Each staining procedure should be briefly explained in the "Materials and methods" section.
6. The wounds were induced in a width of 1 cm, as mentioned in the "Materials and methods" section (on page 3, line 120). According to the results, the wounds are more than 1 cm wide after one week of treatment. The authors should discuss and justify this in the "Discussion" section.
7. On page 7, line 258-261, "Statistically, it was found that the … especially in the rASC/dWJ group," the statement should be corrected. The experimental groups should be compared to the control group.
8. According to Figure 5, the dWJ group (figure 5 (B)) appears to have more CD31+ and LYVE-1+ cells than the control and rASC/dWJ groups. The results should be double-checked and compared to the text.
9. Figure 5 (E) should be defined in the caption.
10. The role of angiogenesis in the wound healing process (for the reference: https://jnanobiotechnology.biomedcentral.com/articles/10.1186/s12951-020-00755-7), as well as the role of rASC/dWJ in promoting angiogenesis during the wound healing process, should be discussed in the "Discussion" section.
11. The "Conclusion" section is poorly written. The main idea that runs through the entire research, as well as the aim of the study, should be highlighted. The research findings should be included, followed by a clear conclusion statement.
Other minor revisions:
- "Wound healing" could be considered as a keyword.
- All "ex vivo," "in vitro," and "in vivo" should be written in italics throughout the manuscript.
- On page 4, line 142, the ImageJ software version should be included.
- The quality of Figure 2 (A) is poor.
- On page 6, the unit for the width of the wounds has not been specified in some cases.
Author Response
- In the "Abstract," the authors should include a background statement on the effect of lymphedema on delayed wound healing.
Answer: Thanks for your suggestion. The Abstract was revised. ----Line 23
Please refer to the answer of Question 2
- The results of in vitrotests (i.e., survival and proliferation of rASCs) should be mentioned in the "Abstract."
Answer: Thanks for your suggestion. The Abstract was revised and results of in vitro tests was added to Abstract (Blue words). .----Line 26~~27.
Abstract: Lymphedema causes tissue swelling due to the accumulation of lymphatic fluid in tis-sue, which delays the process of wound healing. Developing effective treatment options of lymphedema is still an urgent issue. In this study, we aim to fabricate tissue-engineered moist wound dressings with adipose stem cells (ASCs) and decellularized Wharton’s jelly (dWJ) from the human umbilical cord in order to ameliorate lymphedema. The rat ASCs proliferated and an apparent layer was visualized on dWJ at day 7 and 14. A rat tail lymphedema model was developed to evaluate the efficacy of the treatment. Approximately one cm of skin near the base of the rat tail was circularly excised. The wounds were treated by sec-ondary healing (control) (n=5), decellularized Wharton’s jelly (n=5) and ASC-seeded dWJ (n=5). The wound healing rate and the tail volume were recorded once a week from week one to week five. Angiogenesis and lymphangiogenesis were assessed by immunochemistry staining with anti-CD31 and anti-LYVE1. The results showed that the wound healing rate was faster and the tail volume was lesser in the ASC-seeded dWJ group than in the control group. More CD31+ and LYVE-1+ cells were observed at the wound healing area in the ASC-seeded dWJ group than in the control group. This proves that tissue-engineered moist wound dressings can accelerate wound healing and reduce lymphedema by promoting angiogenesis and lymphangiogenesis. ”
- The wound healing process (for the reference: https://www.sciencedirect.com/science/article/pii/S0141813021021541), as well as the relationship between lymphedema and delayed wound healing (for the reference: https://www.liebertpub.com/doi/abs/10.1089/wound.2018.0871), should be explained in the "Introduction" section.
Answer: Thanks for your suggestion. The description of The wound healing process, the relationship between lymphedema and delayed wound healing and the reference were appended in the "Introduction" section.—Line55~56
The two references suggested by Reviewer were listed in Reference 8 and 9.-Line 399~404
- Could the authors elaborate on the evaluation of the differentiation potential of ASCs in the "Materials and methods" section?
Answer: Thanks for your suggestion. The description of the differentiation potential of ASCs was written in the Section of "Materials and method” (-Line 168~172) and Result (Line 226~228).
- Each staining procedure should be briefly explained in the "Materials and methods" section.
Answer: Thanks for your opinion. We add the description as below.
The rats were sacrificed and the wound areas were harvested. All samples were fixed in10% (v/v) neutral-buffered formalin solution (Leica, USA) for 24 h, processed paraffin blocked and sectioned by 5μm thickness. Immunohistochemistry staining were preformed to evaluate angiogenesis and lymphangiogenesis with anti-CD31 (Abcam, UK,), anti-LYVE1 (Invitrogen, USA). Immunohistochemistry was performed on Bond-MAX system (Leica Biosystems, Melbourne, Australia) which was an automated IHC staining system with the Bond Polymer Refine Detection Kit (Leica, UK) according to the manufacture’s instruction. The tissue sections were pretreated using heat-mediated antigen retrieval with sodium citrate buffer (pH 6, epitope retrieval solution 1) for 30 min. Then, the samples were incubated with anti-CD31and anti-LYVE1 at 1:100 dilution for 60 min at room temperature, followed by secondary HRP antibodies. After blocking with peroxide for 5 min, DAB was applied as the chromogen on the stained sections. Finally, hematoxylin for nuclear counter staining was processed, and slides were mounted with DPX (Sigma-Aldrich, MO USA). Images were captured under the bright field of a ZEISS inverted microscope (Zeiss,Germany). – Line 202 to 216.
- The wounds were induced in a width of 1 cm, as mentioned in the "Materials and methods" section (on page 3, line 120). According to the results, the wounds are more than 1 cm wide after one week of treatment. The authors should discuss and justify this in the "Discussion" section.
Answer: Thanks for your opinion.
Compared with the wounds of dWJ and ASC/dWJ group, that of control group were more than 1 cm wide after one week of operation. This initial wound enlargement following skin excision due to muscular retraction and lymphedema. in the "Discussion" section. .—Line306~309 .
- On page 7, line 258-261, "Statistically, it was found that the … especially in the rASC/dWJ group," the statement should be corrected. The experimental groups should be compared to the control group.
Answer: Thanks for your opinion. It was corrected in Line 283~290, which is:
Five weeks after surgery, subcutaneous CD31 & LYVE-1 IHC staining results were obtained. It was observed that LYVE-1+ cells formed the lumen of the lymphatic vessels, and there were also free cells in the interstitial space (Figure 5). However, the staining results showed that the luminal endothelial cells exhibited both LYVE-1 and CD31 (Figure 5A ~5C), which was different from past studies. From the data of calculation, it illustrated that the number of cells exhibiting CD31 and LYVE-1 increased in both the dWJ dressing group and the rASC/dWJ group compared to the rASC/dWJ group. There was no significate difference between rASC/dWJ group and control. Nevertheless, in the rASC/dWJ group, there were statistically increase in comparison to the control and dWJ groups (P<0.05). (Figure 5D and 5E)
- According to Figure 5, the dWJ group (figure 5 (B)) appears to have more CD31+ and LYVE-1+ cells than the control and rASC/dWJ groups. The results should be double-checked and compared to the text.
Answer: Thanks for your opinion. The images of dWJ (Figure 5B) were change. The results were revised as well in 3.4. Histological outcome of the section of Result. - Line 283~290.
- Figure 5 (E) should be defined in the caption.
Answer: Thanks for your opinion. Figure 5 (E) was explained at the 3.4. Histological outcomes. in Line 283~290.
- The role of angiogenesis in the wound healing process (for the reference: https://jnanobiotechnology.biomedcentral.com/articles/10.1186/s12951-020-00755-7), as well as the role of rASC/dWJ in promoting angiogenesis during the wound healing process, should be discussed in the "Discussion" section.
Answer: Thanks for your opinion. The role of angiogenesis on wound healing and the effect ion of rASC/dWJ on promoting angiogenesis were written in the "Discussion" section.
Besides lymphangiogenesis, angiogenesis was crucial for accelerating the rate and quality of the wound healing process by providing nutrition to the injury [81]. As shown by the use of decellularized biological material with stem cells in our study, the clinical applications of biomaterial-assisted cell therapies may hold great promise in regenerative therapy for wound healing and lymphedema treatment in the future. —Line 352~356.
The reference suggested by Reviewer is added to Reference 81. —Line 559.
- The "Conclusion" section is poorly written. The main idea that runs through the entire research, as well as the aim of the study, should be highlighted. The research findings should be included, followed by a clear conclusion statement.
Answer: Thanks for your opinion. The conclusion has been revised according to your suggestion.
Conclusion
The tissue-engineered moist wound dressings fabricated with ASCs and decellularized Wharton jelly could attenuate lymphedema by promoting lymphatic vessel and capillary formation in a rat tail lymphedema model. It revealed the function of accelerate the wound healing by restoring blood and lymphatic circulation after skin damage. Therefore, this dressing has potential as an advanced dressing for further clinical application. —Line 376~380
Other minor revisions:
- "Wound healing" could be considered as a keyword.
Answer: Thanks for your opinion. "Wound healing" was add to keyword. -- Line 36.
- All "ex vivo," "in vitro," and "in vivo" should be written in italics throughout the manuscript.
Answer: Thanks for your opinionsuggestion. "ex vivo," "in vitro," and "in vivo" has been written in italics.
"ex vivo," has been written in italics.–Line 106, 537
"in vitro," has been written in italics–Line 230, 430, 545
"in vivo" has been written in italics–Line 107, 338, 470
- On page 4, line 142, the ImageJ software version should be included.
Answer: Thanks for your suggestion. It was revised as below:
The ImageJ software (version 1.53t). -- Line 150.
- The quality of Figure 2 (A) is poor.
Answer: Thanks for your suggestion. An image was enlarged and adjusted to higher resolution. It will be applied in JPG file to the Editor.
- On page 6, the unit for the width of the wounds has not been specified in some cases.
Answer: Thanks for your suggestion. The width of the wounds in the three groups were descripted in Line 256~272.
- In the "Abstract," the authors should include a background statement on the effect of lymphedema on delayed wound healing.
Answer: Thanks for your suggestion. The Abstract was revised. ----Line 23
Please refer to the answer of Question 2
- The results of in vitrotests (i.e., survival and proliferation of rASCs) should be mentioned in the "Abstract."
Answer: Thanks for your suggestion. The Abstract was revised and results of in vitro tests was added to Abstract (Blue words). .----Line 26~~27.
Abstract: Lymphedema causes tissue swelling due to the accumulation of lymphatic fluid in tis-sue, which delays the process of wound healing. Developing effective treatment options of lymphedema is still an urgent issue. In this study, we aim to fabricate tissue-engineered moist wound dressings with adipose stem cells (ASCs) and decellularized Wharton’s jelly (dWJ) from the human umbilical cord in order to ameliorate lymphedema. The rat ASCs proliferated and an apparent layer was visualized on dWJ at day 7 and 14. A rat tail lymphedema model was developed to evaluate the efficacy of the treatment. Approximately one cm of skin near the base of the rat tail was circularly excised. The wounds were treated by sec-ondary healing (control) (n=5), decellularized Wharton’s jelly (n=5) and ASC-seeded dWJ (n=5). The wound healing rate and the tail volume were recorded once a week from week one to week five. Angiogenesis and lymphangiogenesis were assessed by immunochemistry staining with anti-CD31 and anti-LYVE1. The results showed that the wound healing rate was faster and the tail volume was lesser in the ASC-seeded dWJ group than in the control group. More CD31+ and LYVE-1+ cells were observed at the wound healing area in the ASC-seeded dWJ group than in the control group. This proves that tissue-engineered moist wound dressings can accelerate wound healing and reduce lymphedema by promoting angiogenesis and lymphangiogenesis. ”
- The wound healing process (for the reference: https://www.sciencedirect.com/science/article/pii/S0141813021021541), as well as the relationship between lymphedema and delayed wound healing (for the reference: https://www.liebertpub.com/doi/abs/10.1089/wound.2018.0871), should be explained in the "Introduction" section.
Answer: Thanks for your suggestion. The description of The wound healing process, the relationship between lymphedema and delayed wound healing and the reference were appended in the "Introduction" section.—Line55~56
The two references suggested by Reviewer were listed in Reference 8 and 9.-Line 399~404
- Could the authors elaborate on the evaluation of the differentiation potential of ASCs in the "Materials and methods" section?
Answer: Thanks for your suggestion. The description of the differentiation potential of ASCs was written in the Section of "Materials and method” (-Line 168~172) and Result (Line 226~228).
- Each staining procedure should be briefly explained in the "Materials and methods" section.
Answer: Thanks for your opinion. We add the description as below.
The rats were sacrificed and the wound areas were harvested. All samples were fixed in10% (v/v) neutral-buffered formalin solution (Leica, USA) for 24 h, processed paraffin blocked and sectioned by 5μm thickness. Immunohistochemistry staining were preformed to evaluate angiogenesis and lymphangiogenesis with anti-CD31 (Abcam, UK,), anti-LYVE1 (Invitrogen, USA). Immunohistochemistry was performed on Bond-MAX system (Leica Biosystems, Melbourne, Australia) which was an automated IHC staining system with the Bond Polymer Refine Detection Kit (Leica, UK) according to the manufacture’s instruction. The tissue sections were pretreated using heat-mediated antigen retrieval with sodium citrate buffer (pH 6, epitope retrieval solution 1) for 30 min. Then, the samples were incubated with anti-CD31and anti-LYVE1 at 1:100 dilution for 60 min at room temperature, followed by secondary HRP antibodies. After blocking with peroxide for 5 min, DAB was applied as the chromogen on the stained sections. Finally, hematoxylin for nuclear counter staining was processed, and slides were mounted with DPX (Sigma-Aldrich, MO USA). Images were captured under the bright field of a ZEISS inverted microscope (Zeiss,Germany). – Line 202 to 216.
- The wounds were induced in a width of 1 cm, as mentioned in the "Materials and methods" section (on page 3, line 120). According to the results, the wounds are more than 1 cm wide after one week of treatment. The authors should discuss and justify this in the "Discussion" section.
Answer: Thanks for your opinion.
Compared with the wounds of dWJ and ASC/dWJ group, that of control group were more than 1 cm wide after one week of operation. This initial wound enlargement following skin excision due to muscular retraction and lymphedema. in the "Discussion" section. .—Line306~309 .
- On page 7, line 258-261, "Statistically, it was found that the … especially in the rASC/dWJ group," the statement should be corrected. The experimental groups should be compared to the control group.
Answer: Thanks for your opinion. It was corrected in Line 283~290, which is:
Five weeks after surgery, subcutaneous CD31 & LYVE-1 IHC staining results were obtained. It was observed that LYVE-1+ cells formed the lumen of the lymphatic vessels, and there were also free cells in the interstitial space (Figure 5). However, the staining results showed that the luminal endothelial cells exhibited both LYVE-1 and CD31 (Figure 5A ~5C), which was different from past studies. From the data of calculation, it illustrated that the number of cells exhibiting CD31 and LYVE-1 increased in both the dWJ dressing group and the rASC/dWJ group compared to the rASC/dWJ group. There was no significate difference between rASC/dWJ group and control. Nevertheless, in the rASC/dWJ group, there were statistically increase in comparison to the control and dWJ groups (P<0.05). (Figure 5D and 5E)
- According to Figure 5, the dWJ group (figure 5 (B)) appears to have more CD31+ and LYVE-1+ cells than the control and rASC/dWJ groups. The results should be double-checked and compared to the text.
Answer: Thanks for your opinion. The images of dWJ (Figure 5B) were change. The results were revised as well in 3.4. Histological outcome of the section of Result. - Line 283~290.
- Figure 5 (E) should be defined in the caption.
Answer: Thanks for your opinion. Figure 5 (E) was explained at the 3.4. Histological outcomes. in Line 283~290.
- The role of angiogenesis in the wound healing process (for the reference: https://jnanobiotechnology.biomedcentral.com/articles/10.1186/s12951-020-00755-7), as well as the role of rASC/dWJ in promoting angiogenesis during the wound healing process, should be discussed in the "Discussion" section.
Answer: Thanks for your opinion. The role of angiogenesis on wound healing and the effect ion of rASC/dWJ on promoting angiogenesis were written in the "Discussion" section.
Besides lymphangiogenesis, angiogenesis was crucial for accelerating the rate and quality of the wound healing process by providing nutrition to the injury [81]. As shown by the use of decellularized biological material with stem cells in our study, the clinical applications of biomaterial-assisted cell therapies may hold great promise in regenerative therapy for wound healing and lymphedema treatment in the future. —Line 352~356.
The reference suggested by Reviewer is added to Reference 81. —Line 559.
- The "Conclusion" section is poorly written. The main idea that runs through the entire research, as well as the aim of the study, should be highlighted. The research findings should be included, followed by a clear conclusion statement.
Answer: Thanks for your opinion. The conclusion has been revised according to your suggestion.
Conclusion
The tissue-engineered moist wound dressings fabricated with ASCs and decellularized Wharton jelly could attenuate lymphedema by promoting lymphatic vessel and capillary formation in a rat tail lymphedema model. It revealed the function of accelerate the wound healing by restoring blood and lymphatic circulation after skin damage. Therefore, this dressing has potential as an advanced dressing for further clinical application. —Line 376~380
Other minor revisions:
- "Wound healing" could be considered as a keyword.
Answer: Thanks for your opinion. "Wound healing" was add to keyword. -- Line 36.
- All "ex vivo," "in vitro," and "in vivo" should be written in italics throughout the manuscript.
Answer: Thanks for your opinionsuggestion. "ex vivo," "in vitro," and "in vivo" has been written in italics.
"ex vivo," has been written in italics.–Line 106, 537
"in vitro," has been written in italics–Line 230, 430, 545
"in vivo" has been written in italics–Line 107, 338, 470
- On page 4, line 142, the ImageJ software version should be included.
Answer: Thanks for your suggestion. It was revised as below:
The ImageJ software (version 1.53t). -- Line 150.
- The quality of Figure 2 (A) is poor.
Answer: Thanks for your suggestion. An image was enlarged and adjusted to higher resolution. It will be applied in JPG file to the Editor.
- On page 6, the unit for the width of the wounds has not been specified in some cases.
Answer: Thanks for your suggestion. The width of the wounds in the three groups were descripted in Line 256~272.
Please see the attachment in which all Revision is Written with BLUE words.

Reviewer 4 Report
Please see the attachment.

Author Response
- Abstract Line 21-22; The phrase is too strong, because it is not necessarily true that there is no treatment for lymphedema. The reviewer thinks mild expression would be better, for example, “Developing effective treatment options of lymphedema is still an urgent issue.
Answer: Thanks for your suggestion. It was revised according to your suggestion. -Line 23~24
Abstract: Lymphedema causes tissue swelling due to the accumulation of lymphatic fluid in tis-sue, which affects both physical function and overall quality of life. Developing effective treat-ment options of lymphedema is still an urgent issue. In this study, we aim to fabricate tis-sue-engineered moist wound dressings with adipose stem cells (ASCs) and decellularized Wharton’s jelly (dWJ) from the human umbilical cord in order to ameliorate lymphedema. The rat ASCs proliferated and an apparent layer was visualized on dWJ at day 7 and 14. A rat tail lymphedema model was developed to evaluate the efficacy of the treatment. Approximately one cm of skin near the base of the rat tail was circularly excised. The wounds were treated by sec-ondary healing (control) (n=5), decellularized Wharton’s jelly (n=5) and ASC-seeded dWJ (n=5). The wound healing rate and the tail volume were recorded once a week from week one to week five. Angiogenesis and lymphangiogenesis were assessed by immunochemistry staining with anti-CD31 and anti-LYVE1. The results showed that the wound healing rate was faster and the tail volume was lesser in the ASC-seeded dWJ group than in the control group. More CD31+ and LYVE-1+ cells were observed at the wound healing area in the ASC-seeded dWJ group than in the control group. This proves that tissue-engineered moist wound dressings can accelerate wound healing and reduce lymphedema by promoting angiogenesis and lymphangiogenesis. ”
- Introduction Line 60; needs reference. If possible, please cite the following paper. Effect of Postoperative Compression Therapy on the Success of Liposuction in Patients with Advanced Lower Limb Lymphedema. J Clin Med. 2021 Oct 22;10(21):4852.
Answer: Thanks for your suggestion. The reference was added in Introduction Line 60 and Reference 13—Line 65 and 411
- Material Methods Line132; “lymphoma”. Is it “lymphedema”?.
Answer: Thanks for your opinion.
It is a misspelling and has been revised to “lymphedema”.—Line 141
4.Discussion Line 267-269; The sentence needs references, or might be better to be deleted, since it is not important part for discussion.
Answer: Thanks for your opinion. The references of this part were missing while submit. They were added in Line 298~299.
Please see the attachment in which all Revision is Written with BLUE words.

Round 2
Reviewer 1 Report
REVIEW REPORT2 -MANUSCRIPT JFB-2130345 V2 TITLED: A NOVEL DRESSING COMPOSED OF ADIPOSE STEM CELLS AND DECELLUARIZED WHARTON’S JELLY FACILITATED WOUND HEALING AND RELIEF OF 3 LYMPHEDEMA BY ENHANCING ANGIOGENESIS AND LYMPHANGIOGENESIS IN A RAT MODEL
GENERAL
The authors have largely responded to the issues raised earlier, especially in the material and methods section. However, the issue of figure captions was not well addressed. Only figure legend1 has been modified adequately.
Figure 2: Indicate what each component of the figure is showing. What were the stains in B,C,D and F?
Figure 3: Briefly explain in the caption what is happening to the wound and lymphedema in A,B, and C as healing progresses from week 0 through to 5. Week 0 may be misconstrued to mean no week; would you like to rename it Week 1?
Figure 4: What are the units for y-axis in Figure 4 A? Briefly explicate the trends shown in these graphs in your legend.
Figure 5: Do you mean ‘angio-expression’ instead of ango-expression?. Explain briefly in the figure caption what the stained sections are showing, and mention something about the bar graphs.
NB: the manuscript could benefit from some English editing, especially when it comes to spacing of words (highlighted in yellow in the attached PDF). Some in-text citations appear after the full stop.

Author Response
Q1. Figure 2: Indicate what each component of the figure is showing. What were the stains in B,C,D and F?
Ans: Thanks for your instruction.
The texts corresponding to the Figure 2 B, C, D were in a wrong order. The description of Figure 2 B,C,D is amened and the staining of 2F is added as below.
Figure 2. Rat ASC. A: in vitro characterization of ASC markers at passage 7 to 10. The image and characterization of ASCs by FACS; B: the chondrogenic differentiation of ASC by Alcian Blue staing; C: the osteogenic differentiation of ASC by Alizarin Red staining; D: the adipogenic differentiation of ASC by Oil Red O staining; E: the proliferation curve of ASCs on the Petri dish and dWJ; F: DAPI and HE staining of ASC-dWSJ after cells were seeded for 7 days and 14 days. The upper Magnification 200x; scale bar = 5 um.
Q2. Figure 3: Briefly explain in the caption what is happening to the wound and lymphedema in A,B, and C as healing progresses from week 0 through to 5. Week 0 may be misconstrued to mean no week; would you like to rename it Week 1?
Ans: Thanks for your instruction.
Week 0 represented the time when the skin of mouse tails was cut circularly and treated with the dressings in the same time. One week after treatment was named week 1, and so on.
We explain this in the revised Figure legend as below.
Figure 3. Images of rat tail lymphedema and wound healing after dWJ and rASC/dWJ treatment for five weeks after injury. A: images of rat tails of control group from week 0 to week 5; B: images of rat tails of dWJ group from week 0 to week 5; C: images of rat tails of dWJ group from week 0 to week 5. Photographs of Week 0 represent the mouse tails treated without and with dressings just after surgical excision and the wound width for each group was managed as same as possible. During the wound development, lymphedema of control group was the most significant compared with the two treatment groups. The wounds width of rASC/dWJ groups seems smaller than the other two groups from week one to five after treatment. n=5/group.
Q3. Figure 4: What are the units for y-axis in Figure 4 A? Briefly explicate the trends shown in these graphs in your legend.
Ans: Thanks for your instruction.
As we explained in the second paragraph of section 2.1 on line 145-146, “The increase in volume which represents lymphedema was defined as post- vs. pre-operative volumes of the same animal for each week.” --It is the fold of the tail volume increasing compared to week 0. Thus, the units for y-axis in Figure 4 A is fold. We add the unit directly to Figure 4A
Q4. Figure 5: Do you mean ‘angio-expression’ instead of ango-expression?. Explain briefly in the figure caption what the stained sections are showing, and mention something about the bar graphs.
Ans: Thanks for your instruction.
The misspelling of ‘ango-expression’ is corrected to ‘angio-expression’.
The figure caption are revised according to your suggestion.
Figure 5. The angio-expression of CD31 and LYVE-1 in the excision sites after five weeks. Immunohistochemistry of A: control group; B: dWJ group; C: rASC/dWJ group. The results showed that although both CD31 and LYVE-1 positive cells in treating groups ( rASC/dWJ group and dWJ group) were more than control group, rASC/dWJ group contained the most positive cells among these three groups. The red arrows pointed to capillary blood vessels and the green arrows indicated the lymphatic vessels. 40x magnification; Scale bar = 200 um; 400x magnification; scale bar =20 um. D: The qualification of CD31 positive cells; E: The qualification of LYVE-1 positive cells. The statistical data depicted that both CD31 and LYVE-1 positive cells in rASC/dWJ group were significantly more than control and dWJ group. * P<0.05.
NB: the manuscript could benefit from some English editing, especially when it comes to spacing of words (highlighted in yellow in the attached PDF). Some in-text citations appear after the full stop.
Ans: Thanks for your instruction.
We will send this manuscript for further English editing.
Please see the attachment in which all Revisions are written with BLUE words.

Reviewer 2 Report
None
Author Response
Thanks for your instruction.
We will send this manuscript for further English editing.
